# Advances in Sjögren’s Syndrome Dry Eye Diagnostics: Biomarkers and Biomolecules beyond Clinical Symptoms

**DOI:** 10.3390/biom14010080

**Published:** 2024-01-08

**Authors:** Kevin Y. Wu, Olivia Serhan, Anne Faucher, Simon D. Tran

**Affiliations:** 1Division of Ophthalmology, Department of Surgery, University of Sherbrooke, Sherbrooke, QC J1G 2E8, Canada; yang.wu@usherbrooke.ca (K.Y.W.); anne.faucher@usherbrooke.ca (A.F.); 2Faculty of Medicine, University of Sherbrooke, Sherbrooke, QC J1G 2E8, Canada; 3Faculty of Dental Medicine and Oral Health Sciences, McGill University, Montreal, QC H3A 1G1, Canada

**Keywords:** Sjögren’s syndrome dry eye, diagnostic biomarkers, serum analysis, salivary analysis, tear analysis, tear proteomics, exosomal biomarkers, ophthalmology, molecular diagnostics, ocular surface disease

## Abstract

Sjögren’s syndrome dry eye (SSDE) is a subset of Sjögren’s syndrome marked by dry eye symptoms that is distinct from non-Sjögren’s syndrome dry eye (NSSDE). As SSDE can lead to severe complications, its early detection is imperative. However, the differentiation between SSDE and NSSDE remains challenging due to overlapping clinical manifestations. This review endeavors to give a concise overview of the classification, pathophysiology, clinical features and presentation, ocular and systemic complications, clinical diagnosis, and management of SSDE. Despite advancements, limitations in current diagnostic methods underscore the need for novel diagnostic modalities. Thus, the current review examines various diagnostic biomarkers utilized for SSDE identification, encompassing serum, salivary, and tear analyses. Recent advancements in proteomic research and exosomal biomarkers offer promising diagnostic potential. Through a comprehensive literature review spanning from 2016 to 2023, we highlight molecular insights and advanced diagnostic modalities that have the potential to enhance our understanding and diagnosis of SSDE.

## 1. Introduction

Sjögren syndrome (SS) is a prevalent autoimmune disorder affecting a diverse population globally. Characterized by its direct assault on the exocrine glands, SS presents significant ocular and oral challenges that can profoundly disrupt a patient’s daily life. Primary manifestations include dry eyes and dry mouth, which can range from mild discomfort to severe and debilitating symptoms [1]. Early and accurate diagnosis is paramount to avert potential vision-related complications and mitigate the onset of systemic repercussions. Ophthalmologists, at the frontline of patient care, are crucial in identifying and managing Sjögren syndrome dry eye (SSDE) [2].

As the field of medicine progresses, this review endeavors to give a concise overview of the classification, pathophysiology, clinical features and presentation, ocular and systemic complications, clinical diagnosis, and management of SSDE. Despite advancements, limitations in current diagnostic methods underscore the need for novel diagnostic modalities. Thus, this article will cautiously explore potentially promising diagnostic modalities, with a focus on innovations in serum, salivary, and tear analysis, in the pursuit of refining SSDE diagnosis.

## 2. Overview of Sjögren’s Syndrome Dry Eye

### 2.1. Classification

Sjögren’s syndrome can be delineated into two distinct classifications: primary and secondary. Primary SS manifests independently, while secondary SS arises in tandem with other autoimmune diseases. Rheumatoid arthritis and systemic lupus erythematosus frequently co-exist with secondary SS. However, there have been noted associations with other conditions such as Raynaud’s disease, scleroderma, primary biliary cholangitis, and autoimmune hepatitis [1,3].

### 2.2. Prevalence

Sjögren’s syndrome (SS) affects an estimated 0.06% of the global population, with women constituting over 90% of diagnosed cases [4]. Among patients presenting with dry eye disease (DED), approximately 10% are associated with SS [4]. Alarmingly, approximately two-thirds of these SS cases remain undiagnosed, resulting in a median diagnostic delay of up to 10 years [4]. This significant lag in diagnosis can be attributed to several factors: the ubiquity of symptoms like dry eyes and mouth, the non-specific clinical manifestations often associated with SS, and its insidious disease onset. Notably, many individuals with SS-associated DED might be asymptomatic or display only mild symptoms, yet they may have underlying significant ocular inflammation [5]. Current limitations in robust screening tools and definitive protocols for DED assessment exacerbate the challenge. Additionally, the profound impact of SS on a patient’s quality of life may often be underestimated by clinicians, leading to reduced referrals for comprehensive evaluations [6].

### 2.3. Pathogenesis

The underlying pathogenesis of SS is a multifaceted process, combining genetic predisposition with environmental factors [7]. While the precise etiology remains elusive, the development of primary SSDE can be conceptualized through a sequence of four phases (Figure 1).

Initially, a convergence of genetic vulnerabilities with environmental exposures takes place. This interaction subsequently paves the way for the evolution of autoimmunity, representing the second phase [7,8]. As the process advances, there is a marked degradation of the lacrimal glands, culminating in an aqueous tear insufficiency, constituting the third phase [2].

With the onset of ocular surface hyperosmolarity, pro-inflammatory cytokines become substantially prominent, influencing the entirety of the functional lacrimal unit [2]. It is imperative to understand the multifaceted implications at this juncture:Meibomian gland dysfunction coupled with subsequent or eventual atrophy diminishes the tear film’s lipid layer, escalating evaporation.The mucin layer of the tear film is compromised due to goblet cell apoptosis, resulting in reduced wettability.The degeneration of epithelial cells, on the other hand, causes a loss of microvilli, further exacerbating the wettability issue.A neurogenic inflammation of the lacrimal gland further accentuates the aqueous tear deficiency.Moreover, corneal nerve disturbances diminish corneal sensation and the blinking reflex.

Figure 2 depicts the tear film layers previously mentioned, overlaying the cornea.

This intricate network of events culminates in an unstable ocular surface tear film, which, in turn, reinforces the ocular surface hyperosmolarity. This perpetual interaction drives the relentless cycle of chronic ocular surface inflammation [7].

Progressively, this cascades into the dysfunction of the entire functional lacrimal unit, propelling a self-perpetuating cycle of DED. As the disease reaches its advanced stages, the clinical distinction of SSDE from other dry eye variants becomes intricate. The challenge arises due to symptomatological overlap with general dry eye conditions and pronounced lacrimal unit damage in SSDE, akin to other severe dry eye pathologies [1].

### 2.4. Clinical Manifestations

#### 2.4.1. Ocular Manifestations of SS

The ocular surface disturbances seen in SS present a spectrum of symptoms. Patients may report blurred vision, a sensation akin to a foreign body, ocular burning, light sensitivity (photophobia), and erythema of the eyes. Such manifestations often amplify with extended visual tasks and are exacerbated in environments with low humidity or frigid conditions [5]. A subset of patients, despite evident ocular inflammation, might exhibit minimal-to-no symptoms [5].

Upon clinical examination, diminished tear meniscus and shortened tear breakup time are the typical findings, with conjunctival hyperemia noted in some cases. Vital dye staining using rose bengal or lissamine green uncovers devitalized conjunctival epithelium, while fluorescein highlights punctate epithelial erosions on the corneal surface [5]. It is noteworthy that in SSDE, symptomatic severity does not consistently mirror the physical signs. Advanced stages of SSDE might display ocular surface debris like secretions, filamentary keratitis, and mucous plaques [5]. Occurrences of corneal and conjunctival keratinization, as well as corneal calcification, are sporadically seen [5]. While SSDE primarily stems from an aqueous tear deficit, the concomitant onset of meibomian gland dysfunction and evaporative tear loss, marked by palpebral anomalies like telangiectasias and meibomian gland atrophy, further complicate its differentiation from non-SSDE presentations [9].

#### 2.4.2. Glandular Manifestations of SS

Xerostomia, characterized by oral dryness, emerges as the hallmark glandular symptom of SS, complicating tasks like mastication and phonation. This predisposes individuals to dental caries, oral mycoses, and chronic esophagitis [1]. A frequent clinical observation is the enlargement of salivary glands attributable to sialadenitis stemming from lymphocytic infiltration. Contrarily, lacrimal gland enlargement is infrequent in SS. When present, it necessitates a thorough evaluation to exclude other infiltrative pathologies like lymphoma, sarcoidosis, amyloidosis, and IgG4-related disease [10].

## 3. Current Diagnostic Challenges in Clinical Settings

Sjögren’s syndrome (SS) affects a minor 0.06% of the global population, but its implications are far-reaching [4]. Yet, a concerning statistic is the underdiagnosis—a staggering two-thirds go undiagnosed, with a median delay in diagnosis reaching 10 years [4]. The reasons behind this gap include:The high prevalence of symptoms like dry eye and mouth often masks the underlying SS, posing a challenge for clinicians [4].The clinical manifestations of SS are broad and non-specific, coupled with its insidious onset [11].A fraction of SSDE patients can exhibit no symptoms or just mild ones despite significant ocular inflammation [11].The paucity of dependable screening tools to discern which DED patients should undergo SS evaluation [4].The tendency among ophthalmologists to downplay SS’s significance, leading to fewer referrals for an SS workup [12].

### 3.1. Significance of Timely SS Diagnosis and Its Extraglandular and Systemic Implications

Failure to address SSDE in its early stages could pave the way for vision-threatening complications. These can manifest as neurotrophic keratitis, which in turn could lead to corneal issues ranging from thinning and perforation [5,13]. Moreover, SS is linked to type III hypersensitivity reactions, leading to immune-complex depositions. Moreover, type III hypersensitivity reactions linked to SS could culminate in immune-complex depositions, causing severe inflammation in other ocular structures, including scleritis [14], uveitis [15], optic neuropathy [16], and retinal vasculitis [17,18].

While SS may appear benign initially, systemic complications arise in approximately one-third of patients, affecting their quality of life and posing a serious risk to their health [19]. Notably, approximately half of SS patients report arthralgia. Pulmonary afflictions, such as bronchiolitis or interstitial lung diseases, are seen in 10–20% of SS patients. There are sporadic reports of neurological complications, including sensory neuropathy, myelitis, and meningitis. Hematological manifestations also vary, with conditions such as anemia, leukopenia, thrombocytopenia, and hypergammaglobulinemia seen in 2.8%, 12%, 5%, and 36% of SS patients, respectively [5]. Concurrent conditions like Hashimoto’s thyroiditis, cardiovascular disease, and depression have been noted in 35%, 30%, and 33% of SS patients, respectively. A particularly alarming association is the amplified risk of malignancies in SS patients. The relative risk for non-Hodgkin lymphoma is augmented to 13.8 times, whereas the risk for all malignant and thyroid tumours are increased to 1.5 and 2.6 times, respectively [5]. Consequently, early identification of SS and its ocular and systemic implications are of paramount importance.

The presence of SSDE can cast a significant impact on subsequent ocular surgical choices. In particular, patients with SSDE warrant careful consideration before undergoing refractive surgery or blepharoplasty [20,21,22].

For clinical assessment, tools like the ESSDAI (European Sjögren’s Syndrome Disease Activity Index) offer an insight into the disease’s activity, damage, and manifestations, while the ESSPRI (European Sjögren’s Syndrome Patient Reported Index) serves to evaluate the disease’s impact on a patient’s quality of life and functionality [23].

### 3.2. Contemporary Diagnostic Protocols

In the quest for a more definitive and objective diagnostic tool, the criteria delineated by the American College of Rheumatology/European League Against Rheumatism in 2016 (i.e., 2016 ACR-EULAR Classification Criteria) for primary Sjögren’s syndrome stands as a cornerstone [24]. This system not only delineates the specific criteria but also expounds upon the diagnostic scoring for primary SS. The diagnostic scoring for primary SS is as follows:A labial salivary gland biopsy displaying focal lymphocytic sialadenitis and a focus score of 1 or more foci per 4 mm^2^ is awarded 3 points.The presence of autoantibodies, notably anti-Ro or anti-La, is attributed 3 points.An ocular staining score of 5 or higher, or a van Bijsterveld score of 4 or higher in at least one eye, receives 1 point.A Schirmer’s test result of 5 mm/5 min or lower in at least one eye earns 1 point.An unstimulated whole saliva flow rate of 0.1 mL/min or lower is given 1 point.

For a visual representation of the 2016 ACR-EULAR Classification Criteria, please refer to Figure 3.

Yet, the 2016 ACR-EULAR Classification Criteria underscore the ocular conditions in SS, and there is an evident tilt favoring the labial salivary gland biopsy and positive SS antibody over ocular examinations [24]. Several limitations plague the current diagnostic criteria, including:Serological markers, including anti-RO/SSA, anti-La/SSB antibodies, ANA, and rheumatoid factor (RF), present limitations in SS screening due to their suboptimal sensitivity. Notably, ANA titers demonstrate approximately 80% reliability for SS [25], with a peak sensitivity of 68.3% [26]. Rheumatoid factor manifests in approximately 51% of SS patients [27], with a recorded sensitivity of 53% [28]. Additionally, during the disease’s initial stages, these autoantibodies may remain undetectable. Studies have indicated sensitivity ranges of 69–77% for anti-RO/SSA and 39–44% for anti-La/SSB antibodies [29].Both the saliva flow rate and Schirmer’s test are infrequently administered in clinical environments. Their specificity for SS is debated [9], and they are time-consuming and often deemed impractical for busy ophthalmological and dental practices.The utilization of the van Bijsterveld score (VBS) and ocular staining score (OSS) systems is limited among ophthalmologists. One significant concern is the use of rose bengal, which, when available in strip form, demonstrates limited effectiveness. Moreover, while rose bengal can be compounded into drops, this formulation is not only challenging to procure but also causes considerable discomfort to patients. These issues, combined with the scarcity of lissamine green in numerous eye clinics, further restrict the widespread utilization of these scoring systems. Additionally, a high OSS does not exclusively indicate SSDE, as it can also be observed in NSSDE [9].The minor salivary gland biopsy demonstrates low sensitivity for early SS. Even though most patients tolerate this procedure, complications such as hemorrhage, infection, paresthesia, and mucocele formation can occur [30].

This landscape underscores the pressing need for robust, practical, and cost-effective diagnostic tools for the early and accurate detection of SS.

### 3.3. Rheumatologic Workup

In instances where SS is strongly suspected clinically, a thorough rheumatologic assessment often encompasses a series of autoantibodies, such as:The anti-Ro/SSA and anti-La/SSB, which are hallmarks of SS [31].The potential presence of rheumatoid factor and antinuclear antibodies (ANAs) [28].While anticitrullinated peptide antibodies (ACPAs) are typically linked to rheumatoid arthritis, they can be detected in approximately 10% of SS patients [32].Centromere antibodies (ACAs) appear in an estimated 7% of individuals with SS and have associations with Raynaud disease and scleroderma [33].Antimitochondrial antibodies (AMAs), found in approximately 7% of patients, correlate with primary biliary cholangitis [34].

### 3.4. Advanced Imaging Modalities

#### 3.4.1. In Vivo Confocal Microscopy (IVCM)

The cornea is a stratified structure, being composed of five distinct layers: the epithelium, Bowman’s layer, stroma, Descemet’s membrane, and endothelium. In vivo confocal microscopy (IVCM) is an imaging tool that delves into these layers, with particular utility on visualizing the subbasal nerve network, which nestles between the epithelium and the anterior stroma (as depicted in Figure 4). These corneal nerves are instrumental for epithelial maintenance, corneal sensory perception, and wound recuperation. Additionally, IVCM provides insights into the meibomian and lacrimal glands’ morphology and integrity. The inflammatory repercussions of SS distinctly compromise the corneal subbasal nerve meshwork. Notably, a 2022 retrospective analysis involving 71 participants revealed that, in SS patients, there was a noticeable decline in corneal sensitivity and subbasal plexus density. There was also a discernible increase in nerve tortuosity, inflammatory cell count, and subbasal neuromas, which was particularly pronounced in patients manifesting small fiber neuropathy, when juxtaposed against healthy controls [35]. Despite its diagnostic potential, IVCM is not widely available, and its effective use requires a trained technician for detailed analysis, posing limitations to its accessibility and utility in clinical practice.

#### 3.4.2. Meibography

Focusing on the in vivo visualization of the meibomian glands situated within the eyelids, meibography employs adaptive transillumination and interferometry for image capture. These images can be benchmarked against clinical metrics, allowing the evaluation of the meibomian gland’s structural integrity and morphology. Dry eye disease (DED) resulting from meibomian gland dystrophy can co-exist with the reduced tear secretion observed in Sjögren’s syndrome. A pivotal 2018 study by Kang et al. discerned a more profound meibomian gland dysfunction in SS-associated DED (SSDE) compared to non-SSDE, as evidenced by deteriorated meiboscores and gland expressibility [11]. Another significant 2021 research study with 108 female eyes found a potent correlation between pSSDED duration and meibomian gland atrophy, reinforcing its diagnostic utility [36]. However, the interpretation of meibography results must be approached with caution, as meibomian gland atrophy occurs naturally with aging, potentially confounding the diagnosis. This underscores the need for establishing age-adjusted meibography standards to improve its specificity and reliability in early SS detection.

#### 3.4.3. Imaging of the Lacrimal Gland (MRI/CT/US)

In specific scenarios, imaging modalities like magnetic resonance imaging (MRI), computerized tomography (CT), or ultrasound (US) can be employed to scrutinize the lacrimal gland’s structure and functionality. In the context of pSS, salivary gland ultrasonography has garnered attention for its diagnostic prowess, monitoring of anatomical shifts, therapeutic responses, and disease activity [37]. Moreover, MRI sialography is regarded as the benchmark for disease stage assessment. A trailblazing 2022 study presented a novel approach using radiomics—a sophisticated analysis of radiologic images—to discern textual differences between pSS-afflicted and healthy lacrimal glands [38]. In pSS manifestations, the lacrimal gland’s bifurcated lobes undergo changes, producing a modified MRI signature distinctly visible in T-1 weighted coronal images with fat saturation.

While the aforementioned advanced imaging modalities hold promise for offering detailed insights into ocular manifestations of diseases like Sjögren’s syndrome, their adoption in routine clinical practice remains limited. Several pertinent challenges hinder their widespread acceptance: the absence of extensive multicentric prospective studies to validate their efficacy, an absence of comprehensive cost-effectiveness analyses, and a notable gap in evidence elucidating their capability to differentiate between SSDE and NSSDE (non-SSDE). Consequently, the clinical utility of these imaging techniques remains uncertain. Without robust data to underscore their potential impact on patient care, their integration into everyday clinical decision-making processes is tentative at best. For these tools to gain traction, rigorous research efforts that directly address these lacunae are imperative.

## 4. Novel Diagnostic Methods

There remains a complexity in diagnosing Sjögren’s syndrome due to its array of signs and symptoms that often overlap with other inflammatory diseases [39]. Current diagnostic tools such as serum antibody testing often yield high rates of false negative results [40] and are time consuming [41]. The diagnosis of SS can be delayed by as much as 6.5 years from symptom onset, leading to inappropriate disease management and the emergence of complications [42]. In fact, certain treatments such as biological agents are known to be more beneficial if started within the first 5 years of disease [43]. Moreover, the present non-invasive diagnostic methods lack the necessary specificity and sensitivity, resulting in SS diagnosis involving invasive gland biopsy and functional assessments [44]. As a result, many have researched alternative non-invasive and disease-specific diagnostic tools to provide a quicker diagnostic while enabling treatment monitoring [41,45].

### 4.1. Serum Analysis

#### 4.1.1. Serum Proteomics

Novel serum biomarkers and proteins have been identified as promising candidates for SS diagnosis [46]. A biomarker, as defined by Jonsson et al., is a quantifiable and objectively assessable substance that functions as an indicator of typical or abnormal biological processes [47]. In an exploratory study analyzing serum biomarkers as a potential method for discriminating SS from rheumatoid arthritis (RA) and systemic lupus erythematosus (SLE), four biomarkers were found to distinguish SS from these other autoimmune diseases. This study involved an analysis of 63 different serum biomarkers. Findings showed that brain-derived neurotrophic factor (BDNF) and inducible T-cell alpha chemoattractant (I-TAC) CXCL11 significantly differentiated between SS and RA patients. A low BDNF concentration and a high I-TAC/CXCL11 concentration were found to significantly correlate with SS diagnosis compared to RA. On the other hand, sCD163 and Fractalkine/CX3CL1 biomarkers were found to significantly differentiate SS from SLE patients. They found that a low concentration of sCD163 and Fractalkine/CX3CL1 significantly increased the probability of having an SS diagnosis compared to SLE. However, there were no biomarkers that could distinguish SS from both RA and SLE. When determining sensitivity and specificity scores for different combinations of these four biomarkers, it was established that low BDNF and high Fractalkine concentrations had a specificity score of 92.6%, but a lower sensitivity score of 19% for SS. When combining a high concentration of I-TAC and low sCD163, there was a specificity score of 98.1% and a low sensitivity score of 7.1% for SS [46].

The utilization of immunoaffinity mass spectrometry to study and quantify the presence of specific serum proteins has evolved into a promising technique for identifying biomarkers for SS diagnosis. The levels of CXCL10, a chemokine involved in immunological diseases like SS and SLE, were compared in human serum samples from SS and healthy patients via mass spectrometry and ELISA analysis. The results demonstrated a significantly higher amount of CXCL10 in SS patient serum samples compared to healthy samples. To further support this finding, the levels of DPP4, a protease responsible for truncating CXCL10, were significantly lower in SS patients compared to healthy patients. This revealed a reduced ability for DPP4 to cleave CXCL10, which ultimately explained the increased levels of CXCL10 in SS patients. This study not only demonstrated a use for immunoaffinity targeted mass spectrometry as a means of analyzing serum proteomics, but also CXCL10 as a biomarker for SS diagnosis [48].

Proteomic fingerprinting, a method that analyzes protein patterns and profiles in samples such a serum or saliva, holds considerable promise for SS diagnosis. A study analyzing serum samples found four protein peaks that were potential biomarkers for SS. They used a combined application of weak cationic exchange (WCX) magnetic beads and MALDI-TOF-MS (matrix-assisted laser desorption/ionization time-of-flight mass spectrometry) to identify specific proteomic peaks. It was found that protein peaks of m/z at 8133.3 and 4837.3 positively correlated with SS, while protein peaks of m/z at 2220.1 and 11,972.4 negatively correlated with SS. This study did not identify the protein components of these peaks, as it was not required to establish this diagnostic model, but set out to characterize the components of these peaks in a future study. Nonetheless, this presented a novel diagnostic method for SS through proteomic fingerprinting [49].

In a study comparing tear miRNA expression in 18 SS and 8 healthy control participants, 43 different miRNAs were measured with real-time polymerase chain reaction (rt-PCR). There were four miRNAs that were found to be significantly higher (miR-16-5o, miR-34a-5p, miR-142-3p, and miR-223-3p), while 10 miRNAs were significantly lower (miR-30b-5p, miR-30c-5p, miR-30d-5p, miR-92a-3p, miR-134-5p, miR-137, miR-302d-5p, miR-365b-3p, miR-374c-5p, and miR-487b-3p) in tears samples from the SS group compared to the healthy group. The authors of the study suggested that these miRNAs might be involved in SS pathogenesis [50].

#### 4.1.2. Novel Candidate Serum Antibody

In a study conducted as part of the Dry Eye Assessment and Management (DREAM) study, researchers set out to identify if new serum autoantibodies were associated with DED. They found that SS patients exhibited a higher prevalence (33%) of salivary protein-1 (SP-1) autoantibody compared to those without SS or with other autoimmune diseases (19%). There was no important difference in the prevalence of parotid secretory protein (PSP) and carbonic anhydrase 6 (CA-6) between SS samples (9.4% and 21%, respectively) and non-SS samples (13.5% and 15%, respectively) [51]. These findings shed light on SP-1 as a potential marker for SS.

#### 4.1.3. Serum Exosomes

Exosomes, which are a subtype of extracellular vesicles released by cells, are tiny sacs that contain proteins and nucleic acids. They participate in various functions, such as intercellular signaling and biological processes, and can be found in a variety of bodily fluids such as serum, saliva, and tear [52]. They carry small RNA, known as micro-RNA, that maintain cellular homeostasis. Exosomes are believed to hold a role in immune tolerance and alterations in exosomal miRNA and thus are important in the development of diseases [53]. Exosomes have been introduced recently as potential biomarkers for SS diagnosis. In a study assessing serum exosomal RNA in mice with early-to-intermediate SS, there were 5 miRNAs (miRNA-127-3p, miRNA-409-3p, miRNA-410-3p, miRNA-541-5p, and miRNA-540-5p) that were statistically significantly increased in this mice cohort, which could serve as biomarkers of SS. This study laid the groundwork to understand the possible role of miRNA in inflammatory diseases like SS. Nonetheless, future research conducted on human serum is imperative to confirm if there are similar correlations between exosomal miRNA and SS [53].

#### 4.1.4. Androgen Deficiency

In a study involving a metabolomic analysis of 222 patients’ serums, it was identified that a decrease in serum androgens was significantly associated with DED. Androgens are known to have an impact on lacrimal and meibomian gland function. Therefore, a decrease in serum androgens can result in a decrease in tear volume, the stability of tear film, and hyperosmolarity [54]. Moreover, another demonstration of androgen deficiency potentially playing a role in SS was an analysis involving serum examination in women with SS. Their results demonstrated that SS-diagnosed women are androgen-deficient; more specifically, they are deficient in androstane-3beta, 17beta-diol, dehydroepiandrosterone (DHEA), dihydrotestosterone (DHT), androsterone-glucuronide (ADT-G), androstane-3a, and 17beta-diol-G compared to controls, with the study concluding that this hormonal imbalance can be related to SS [55]. As a result, androgens may be a potential treatment for DED in SS. There are a few studies that have explored this pathway for treatment. For example, Bizzaro et al. demonstrated that three patients with DED and SS who underwent 60 days of oral testosterone treatment had a substantially increased Schirmer’s tests in contrast to patients who received the placebo treatment for the same duration [56]. Golebiowski et al. conducted a randomized placebo-controlled study on 40 postmenopausal women with DED utilizing transdermal testosterone or estrogen or both and compared to a control group, there was no beneficial effect of this treatment apart from a positive effect on tear secretion in patients who used both the testosterone cream and estrogen gel [57]. Therefore, the effects of androgen treatment remain to be conclusively established in a larger cohort study for DED in SS, but the association between androgens and SS pathophysiology is a promising aspect for future research on therapeutic interventions.

#### 4.1.5. Vitamin D Deficiency

It has been established that Vitamin D deficiency is related to the development of autoimmune disorders including RA, SLE, and multiple sclerosis (MS). In a study assessing serum Vitamin D (25 (OH) D_3_) within a cohort of 74 female participants diagnosed with SS, there was a correlation between a decline in Vitamin D and the manifestation of SS signs; however, this association was not observed with SS symptoms [58]. Radíc et al. found that patients afflicted with SS had lower serum vitamin D concentrations in comparison to healthy controls. Furthermore, they analyzed the ethnic and geographical background of these patients to see if there was an association between these two factors and vitamin D levels, but their analysis did not reveal any correlation between these factors among SS patients [59].

### 4.2. Saliva Analysis

#### 4.2.1. Salivary Proteomics

The analysis of salivary proteomics offers a non-invasive diagnostic approach for the detection of SS. Using liquid chromatography tandem mass spectrometry (LC–MS/MS) to compare salivary proteomics profiles between patients with SS and those with SS-like symptoms not meeting diagnostic criteria, a study revealed distinct salivary proteome composition in SS participants. They discovered that 34 proteins were significantly upregulated in SS participant samples, of which the topmost upregulated proteins were neutrophil elastase, calreticulin, tripartite motif-containing protein 29, clusterin, and vitronectin. These proteins have crucial roles, including acting as inflammatory mediators, participating in pathogen recognition, and regulating host defense pathways. Therefore, these biomarkers could be implicated in SS development and autoimmunity [44]. In another study analyzing saliva proteomics, a total of 38 proteins were significantly elevated in the SS group compared to the control group. They found that the most upregulated proteins of these 38 were neutrophil gelatinase-associated lipocalin (LCN2), granulins (GRN), calmodulin (CALM), epididymal secretory protein 1 (NPC2), and calmodulin-like protein 5 (CALML5), which are proteins involved in innate immunity, cell signaling, and wound repair [60]. Jung et al. noted in their review article that salivary levels of inflammatory cytokines and calprotectin were elevated in SS patients compared to healthy controls and correlated with disease severity, but had limited specificity for SS. Nonetheless, their investigations demonstrated a promising role for these two salivary biomarkers in SS diagnosis [61].

Endogenous proteoglycan 4 (PRG4), a mucin-like glycoprotein secreted by the parotid gland, was identified in saliva following an investigation quantifying PRG4 in tear and saliva samples. The concentration of PRG4 was measured in the saliva study participants. A 2.3-fold increase of PRG4 was identified in SS patients compared to the control group. This statistically significant increase demonstrates the potential use of salivary PRG4 as a biomarker and potential drug target for SS [62].

There is a higher incidence of small airway disease and chronic obstructive pulmonary disease in patients with SS. A diagnostic test for these pulmonary pathologies includes sputum analysis, in which B-cell activating factor (BAFF), interleukin-6 (IL-6), and interleukin-8 (IL-8) are searched for. A study involving sputum analysis revealed that BAFF, IL-6, and IL-8 were significantly increased in SS patient sputum compared to the control, suggesting that they may be involved in the development of associated airway disease in SS. In addition, BAFF was found to significantly correlate with both IL-6 and IL-8, while IL-8 correlated with IL-1. In SS, it is believed that BAFF acts as an important mediator in driving B-cell hyperactivation and differentiation, which aligns with the elucidated findings of this study. Moreover, IL-6 is proposed to play a role in lymphocytic infiltration of the glands and germinal center formation and IL-8 is involved in neutrophil chemotaxis in SS. Therefore, the fact that IL-6 and IL-8 are both elevated in SS patient sputum aligns with their inflammatory roles in SS. These findings underscore the promising non-invasive potential of salivary proteomics as a diagnostic tool for SS [63]. Given the pivotal role that B cells play in the pathogenesis of SS, targeting B cells as treatment of SS through Rituximab, an antibody specific to B-cell membrane protein (anti-CD20), can be a potential avenue for SS management. To date, clinical trials involving Rituximab have demonstrated an improvement of SS symptoms and increased BAFF levels. In addition to B-cell targeted therapy, there is ongoing research exploring interleukin-specific therapy, exemplified by the use of Tocilizumab, an IL-6 receptor antagonist. However, data have yet to be produced for SS-specific treatments [64].

#### 4.2.2. Salivary Exosomes

The first time SS autoantigens were detected in exosomes was in a 2005 study, where it was identified that salivary gland epithelial cells released exosomes that contained autoantigens like anti-SSA, anti-SSB, and Sm. They isolated salivary exosomes from patients with rheumatic disorders via ultracentrifugation and analyzed the contents of these extracellular vesicles via electron microscopy, immunoblotting, or immunoprecipitation. They concluded from their results that exosomes are a pathway in which intracellular autoantigens are presented to autoreactive lymphocytes of the immune system, triggering an autoimmune response and the development of SS [65].

In a 2010 study involving saliva samples from four SS participants and four healthy controls, exosomal miRNA was successfully characterized for the first time by using quantitative PCR and microarray hybridization [66]. In another article, it was found that microRNA patterns distinguished non-SS control patients from SS participants. They found an array of microRNAs that were upregulated in the SS minor salivary gland, therefore demonstrating that microRNA are promising biomarkers of SS disease [67].

To explore salivary extracellular vesicles as potential SS disease biomarkers, researchers used liquid chromatography–mass spectrometry (LC–MS) on salivary samples to separate and purify extracellular vesicles to then identify proteomic biomarker profiles. They found three proteins that were significantly higher in extracellular vesicles of SS participants compared to non-SS sicca participants: CD44, major vault protein (MVP), and neutrophil gelatinase-associated lipocalin (NGAL). Meanwhile, when comparing saliva exosome proteomics in SS participants vs. healthy controls, the three most expressed proteins were Ficolin-1 (FCMN1), CD44, and ANXA4. These proteins play a role in both adaptive immunity through their action on T cells and innate immunity through their mediation of antigen presentation on MHC class 1, which could potentially explain their involvement in the pathogenesis of SS [68].

Another report studying the association between Epstein–Barr virus (EBV) and SS found that there was an increase in EBV-specific microRNA (ebv-miR-BART13-3p) in the salivary glands of patients with SS. This microRNA targeted STIM1, a calcium sensor that is involved in cell signaling and that plays an essential role in salivary gland function. They found that EBV-specific miRNA is transported between cells via exosomes, to then target STIM1. The authors hypothesized that this has an implication in the development of SS and salivary gland dysfunction via the abnormal regulation of calcium signaling [69].

Recently, microarrays and real-time polymerase chain reaction techniques were employed to analyze microRNA found in exosomes derived from mouth rinse samples for SS and healthy control participants. There were 12 miRNAs discerned as potential biomarkers for SS. Among identified miRNAs, there were three that were significantly different between SS and control participants, let-7b-5p, miR-1290, and miR-34a-5p, suggesting their utility in diagnosing SS. Moreover, their research established that the combination of two miRNAs as a diagnostic approach for SS provided a more valid and accurate method of diagnosing SS. For example, when combining miR-1290 and let-7b-5p, results demonstrated a *p*-value of < 0.001 with an area under the curve (AUC) of 0.856, a sensitivity score of 91.7%, and a specificity score of 83.3%. By combining these two miRNAs, they demonstrated a superior diagnostic utility than either miRNA alone [70].

In the 2017 study by Aqrawi et al., among the proteins found in exosomes isolated from a saliva sample, there were five proteins that stood out to be the most upregulated in SS participants compared to healthy control participants: adipocyte plasma membrane-associated protein (APMAP), guanine nucleotide-binding protein subunit alpha-13 (GNA13), WD repeat-containing protein 1 (WDR1), tyrosine-protein phosphatase non-receptor type substrate 1 (SIRPA), and lymphocyte-specific protein 1 (LSP1). These biomarkers play various roles including immune system activation and adipocyte differentiation and are potential candidates for SS diagnosis purposes. However, the authors mentioned that a larger SS cohort was needed to confirm their findings [60].

Over the years, exosomes have gained recognition as therapeutic methods for SS because of their characteristics such as stability, low immunogenicity and toxicity, prolonged half-life, and their ability to cross the blood–brain barrier [71]. Nevertheless, presumed therapeutic findings, such as the use of mesenchymal stem cell-derived exosomes [52], are currently in the experimental stage and significant progress remains before establishing exosome-based treatment for SS [71].

#### 4.2.3. Signaling Pathways

In their 2001 study, Das et al. established that cellular pathway dynamics can also assist in SS diagnosis. In fact, this study identified three pathways that were enriched in saliva samples from SS participants. These pathways were JAK-STAT signaling in response to interleukin-12 (IL-12) stimulation, superoxide metabolic process, and phagocytosis. On the other hand, there were pathways that were enriched in saliva samples from healthy participants, including neutrophil degranulation, negative regulation of peptidase activity, and NABA matrisome-associated. The authors further commented on their findings about the JAK-STAT signaling pathway following IL-12 activation, mentioning that JAK inhibitors have been previously shown to be an alternative treatment in RA by suppressing inflammatory cytokines and chemokines. Therefore, signaling pathway activity has the potential to not only assist in SS diagnosis but to also act as a target for therapeutic intervention [62].

### 4.3. Tear Analysis

#### 4.3.1. Tear Proteomics

Tear proteomic analysis has gained popularity as a diagnostic method for SS via biomarker investigation, as tears can be collected in a non-invasive manner, allowing for easy identification of ocular surface biomarkers [72].

In a 2020 prospective case–control study, researchers studied tear film mucin MUC5AC and its association with different cytokines as a potential diagnostic tool for SS. It was established that participants with SSDE had reduced tear MUC5AC levels compared to NSSDE, which correlated with a higher conjunctival staining score. They also studied the differences in tear cytokines in SS participants compared to non-SS participants. It was established that IL-8 and IL-6 both increased in SS tears when stratified for low MUC5AC. Notably, IL-8 levels were significantly different between SS and non-SS participants. In NSSDE patients who had low MUC5AC, there was an absence of elevated IL-8 in this cohort’s tears. These findings support the use of MUC5AC and IL-8 as prospective tear biomarkers for SS diagnosis [73]. Yu et al. also explored tear mucin correlation with SS. Both MUC5AC and MUC19 were significantly decreased in SS patients compared to individuals without SS, therefore suggesting its use as a diagnostic tool. In addition, they hypothesized that this decrease in eye mucins is the cause for dry eye symptoms in SS, presenting a potential avenue for therapeutic intervention for SS-related symptoms [74].

The inflammatory cytokines that are secreted in SS due to tear film instability stimulate matrix metalloproteinase-9 (MMP-9) synthesis and secretion. This increase in MMP-9 leads to inflammation of the eye and reduced epithelial cells and ocular mucin, leading to SS-associated symptoms and signs. MMP-9 was evaluated as a supplemental diagnostic test for SSDE, and it was found that among patients who tested positive for tear MMP-9, 80% had a Schirmer’s test result of less than 5 mm. This is significantly higher than the 55.6% MMP-9 negative patients who have a similar Schirmer’s test result. MMP-9 was also positive for 63.5% of patients with SS who were experiencing symptoms of dry eyes [75]. Recently, the association between thrombospondin-1 (TSP-1), an inhibitor of MMP-9, and SS was recognized. It was found that a low tear TSP-1/MMP-9 ratio was significantly associated with SS diagnosis, demonstrating the diagnostic value of this ratio to distinguish SS and NSSDE patients [40]. InflammaDry, a modification of lateral flow devices using two antigen-specific antibodies to capture MMP-9 antigens from the tear sample, is a method that can be used to measure MMP-9 in DE patients. This method has a sensitivity score from 66 to 97% in DE patients, demonstrating its utility for SS [76].

In the 2019 paper by Aqrawi et al., three tear proteins were found to be upregulated in SS patients in contrast to non-SS and healthy controls that could be potential biomarkers for this autoimmune disease: LIM domain only protein 7 (LMO7), E3 ubiquitin-protein ligase HUWEI (HUWEI), and tumour protein D52 (TPD52). These are proteins involved in roles such as ubiquitination and B-cell differentiation [68].

Another potential candidate biomarker for SS diagnosis is A Desintegrin and Metalloproteinase domain-containing protein 10 (ADAM10). In a 2022 study, ADAM10 was found for the first time to be significantly increased in SS patients compared to non-SS patients [72].

Through tear protein analysis, lactoferrin (LACTO) and lipocalin-1 (LIPOC-1) were also identified as potential biomarkers for SS. It is known that LIPOC-1 is involved in lipid transport in tears to protect the ocular surface, while LACTO has bacteriostatic, anti-inflammatory, and antioxidant functions. In a study reviewing the charts of 110 participants, lower tear protein levels of LACTO and LIPOC-1 were found in SS patients compared to non-SS. These tear biomarkers demonstrated a higher diagnostic accuracy than current diagnostic methods such as Schirmer’s test, Jones test, tear breakup time (TBUT), corneal staining, and conjunctival staining [77].

Epidermal fatty acid binding protein (E-FABP) was also found to be a promising biomarker for SS. In a prospective study using enzyme-linked immunosorbent assay (ELISA), significantly lower E-FAB concentrations were found in the tears of SS participants. On the other hand, no difference was noted in saliva and serum samples. E-FAB is thought to play a role in the ocular surface barrier, which explains that a decrease in E-FAB can damage this barrier and lead to increased tear evaporation and DED [78].

The above-mentioned biomarkers have the potential to proactively screen patients for SS and could significantly help in the early diagnosis, prognosis, and monitoring of disease.

#### 4.3.2. Signaling Pathways

The 2019 paper by Aqrawi et al. also identified upregulated signaling pathways in SS patient tears compared to non-SS sicca controls. The most amplified pathways were Wnt receptor signaling, MAP kinase cascade, ubiquitination, and tumour necrosis factor (TNF)-mediated signaling. These pathways were involved in ubiquitination and B-cell differentiation, demonstrating an involvement in both innate and adaptive immunity as well as retina homeostasis [68]. Several enriched pathways were identified in SS tears, including leukocyte transendothelial migration, protein–lipid complex remodeling, and collagen catabolic process [62]. It is worth noting, however, that the authors of this study declared that there were some overlapping results in pathways found in SS and healthy control tears [62]. Nonetheless, despite certain overlaps, pathway dynamics in patient tears hold promise as a tool for diagnostic support.

#### 4.3.3. Tear Osmolarity

In recent years, tear osmolarity has been introduced as a diagnostic method for DED. A cross-sectional study involving 155 participants evaluated osmolarity levels in the tears of both cohorts using TearLab Osmolarity System. They found that the mean tear osmolarity in SS participants was 311.1 +/− 16.4 mOsm/L, while it was 297.7 +/− 12.7 mOsm/L in the non-SS DE group. Tear osmolarity values were much higher in the SS group. Additionally, they found a statistically significant positive correlation between mean tear osmolarity and corneal, as well as between conjunctival staining scores (OSS) in SS patients and symptom severity scores (OSDI). There was a negative correlation between mean tear osmolarity and Schirmer’s test [79]. Similar results were established by Kook et al., in which tear osmolarity positively correlated with OSS scores and negatively correlated with TBUT and Schirmer’s test. However, they did not identify a significant correlation with OSDI scores [75].

Versura et al. set out to identify tear osmolarity threshold values for DED diagnosis using the TearLab Osmolarity System. Their investigation unveiled that tear osmolarity normal values were 296.5 +/− 9.8 mOsm/L, while the threshold for dry eye was 305 mOsm/L (AUC 0.737). Moreover, they determined that the cut-off values for moderate and severe dry eye were 309 mOsm/L (AUC 0.759) and 318 mOsm/L (AUC 0.711), respectively [80].

There are some limitations to tear osmolarity measurement, as it can be affected by confounding factors such as air flow, humidity, medication, and seasonal or diurnal variations [79]. In addition, measuring tear osmolarity can be a challenge in SS patients due to their notably limited tear meniscus volume [81]. Therefore, there is a need for improvements in tear osmolarity measurement techniques, but it remains a credible method for diagnosing SS.

#### 4.3.4. Tear Ferning

The evaluation of tear ferning patterns is an alternative to SS diagnosis, as it a cost-effective and simple test to perform to distinguish DE and non-DE patients. In addition, tear ferning has demonstrated high sensitivity (80–90%) and high specificity (75–89%) scores for SS. In healthy eyes, the ferning pattern manifests as dense and uniformly structured ferns, while in DED eyes, tear samples display irregular and shorter ferns that are more widely spaced [82]. In a study evaluating the tear ferning test (TFT) as a potential diagnostic for SS, an abnormal TFT was detected in 90% of samples from primary SS participants and 81% with secondary SS, while only a mere 11% of control samples had an abnormal TFT. The data derived from this study showed that TFT had the highest sensitivity in comparison to other non-invasive tests for DE [83]. There are multiple factors that can affect tear ferning patterns, such as the tear volume, room temperature, humidity during sample drying, and time between collection and assessment [82]. As a result, precautions should be taken to ensure the accuracy and reliability of tear ferning assessments.

#### 4.3.5. Barriers to Clinical Translation: The Gap between Innovation and Implementation

In short, innovative diagnostic methodologies have emerged, holding transformative potential for the timely, non-invasive, and accurate diagnosis of SS. Thanks to advances like immunoaffinity mass spectrometry, the analysis of serum, saliva, and tear proteomics has become more accessible, permitting accurate quantification of specific proteins implicated in SS [45]. Unveiling the underlying pathways and proteins in SS pathogenesis has led to the identification of promising biomarkers, including CXCL10, MUC5AC, the JAK-STAT signaling pathway, MMP-9, and miRNA derived from exosomes. Remarkably, certain proteins, such as LACTO and LIPOC-1, demonstrate superior diagnostic accuracy compared to traditional methods like Schirmer’s test or conjunctival staining.

Yet, the translation of these discoveries into routine clinical practice has been challenging. Despite their potential, there are significant barriers hindering their clinical adoption. The absence of large-scale studies encompassing broad demographic data poses a challenge in determining the broad efficacy and applicability (i.e., external validity) of these methods. There is also a marked lack of cost-effective analyses, making it uncertain if these diagnostic tools would be economically viable in broader healthcare settings. Without robust data on the clinical utility of these methods and whether their application would modify clinical care, their clinical translation remains difficult.

Additionally, tear osmolarity has been identified as a reliable method for diagnosing SS and DED. However, standardizing a precise cut-off for SS hyperosmolarity remains a challenge, given the discrepancies observed across various research publications. Achieving this standardization would provide more definitive guidance for clinicians. The high cost of equipment necessary for tear osmolarity and tear ferning tests also presents a significant barrier to their widespread adoption among ophthalmologists. Tear ferning has also been highlighted as a pivotal diagnostic tool for SS, with studies emphasizing its superior sensitivity for SS relative to other non-invasive techniques.

## 5. Conclusions

The delineation between SSDE and NSSDE, despite their overlapping clinical presentations, remains paramount given the potential severe complications associated with SSDE. The spectrum of diagnostic biomarkers presented in this review, from serum to salivary and tear analyses, offers a fresh perspective into the potential of precise SSDE identification. The burgeoning field of proteomic research and the introduction of exosomal biomarkers indeed hold substantial promise for enhancing SS diagnosis. Yet, as underscored, it is paramount to address the existing translational challenges to ensure that these diagnostic tools not only remain theoretically potent but are also pragmatically applicable in real-world clinical settings. While emphasizing that the importance of proteomic studies as a primary diagnostic avenue for SS is essential, parallel efforts to bridge the translational gaps are equally imperative. Achieving a more rapid and precise SS diagnosis will undoubtedly refine disease management and elevate patient outcomes. This accentuates the ongoing, pressing demand for sophisticated diagnostic tools in the realm of SSDE, as we venture beyond mere clinical symptoms to more nuanced molecular determinants.

## Figures and Tables

**Figure 1 biomolecules-14-00080-f001:**
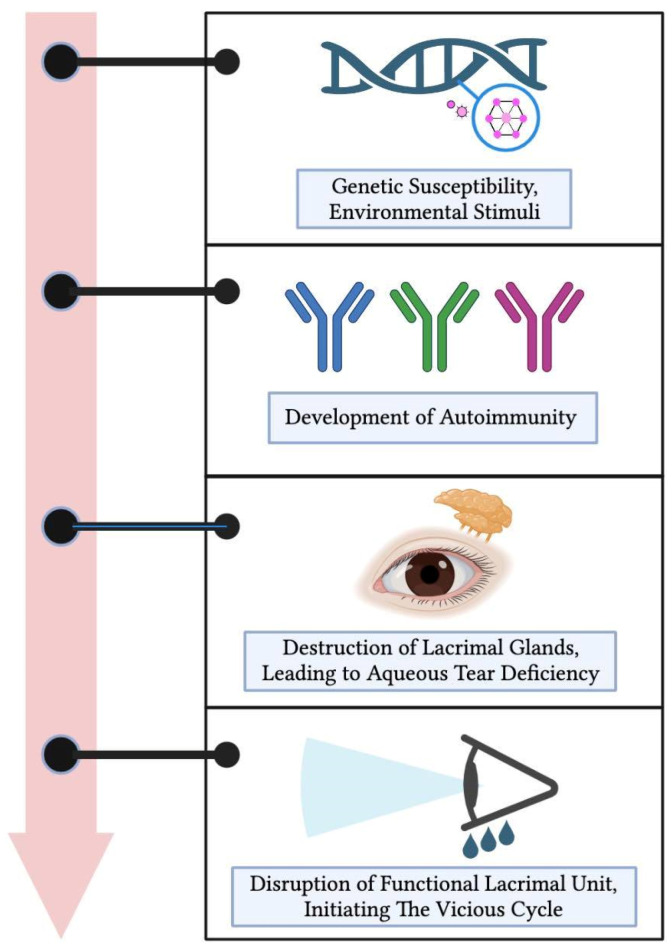
Pathogenesis of primary Sjögren’s syndrome dry eye. Beginning with genetic predispositions and environmental triggers, it progresses through the development of autoimmunity, leading to the destruction of lacrimal glands and a subsequent aqueous tear deficiency. Ultimately, this results in the disruption of the entire functional lacrimal unit, initiating a self-perpetuating vicious cycle of dry eye disease.

**Figure 2 biomolecules-14-00080-f002:**
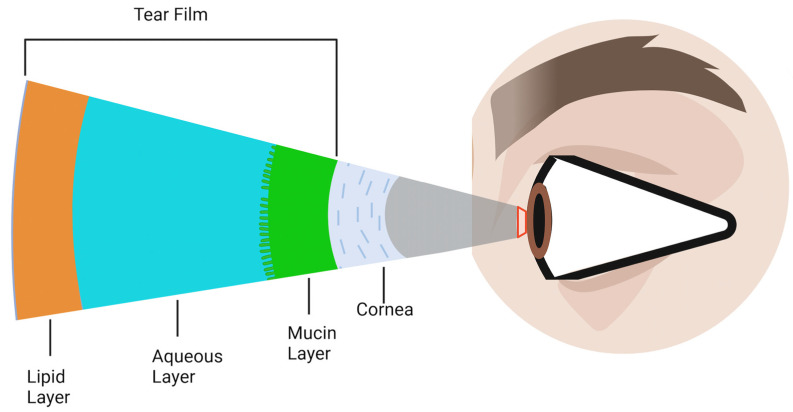
Schematic representation of the tear film layers overlaying the cornea. The tear film is composed of three primary components: the outermost lipid layer, the central aqueous layer, and the innermost mucin layer. Although depicted as distinct layers, in reality, they are not sharply defined but rather merge and overlap into one another.

**Figure 3 biomolecules-14-00080-f003:**
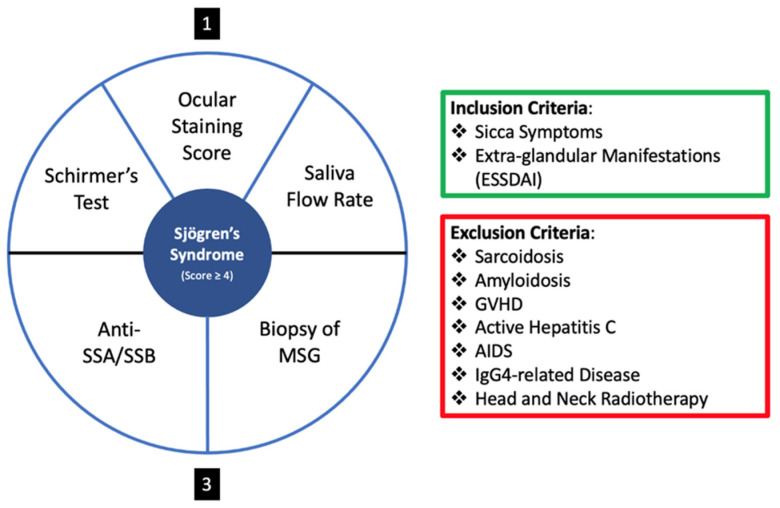
Diagnostic Scoring System of the 2016 ACR-EULAR Classification Criteria for primary Sjögren’s syndrome. The criteria emphasize the importance of labial salivary gland biopsies, autoantibody presence, and various ocular and salivary tests in the evaluation and diagnosis of primary Sjögren’s syndrome. Schirmer’s test, Ocular Staining Score and Saliva Flow Rate give a score of one each, and Anti-SSA/SSB and Biopsy of MSG give a score of 3 each.

**Figure 4 biomolecules-14-00080-f004:**
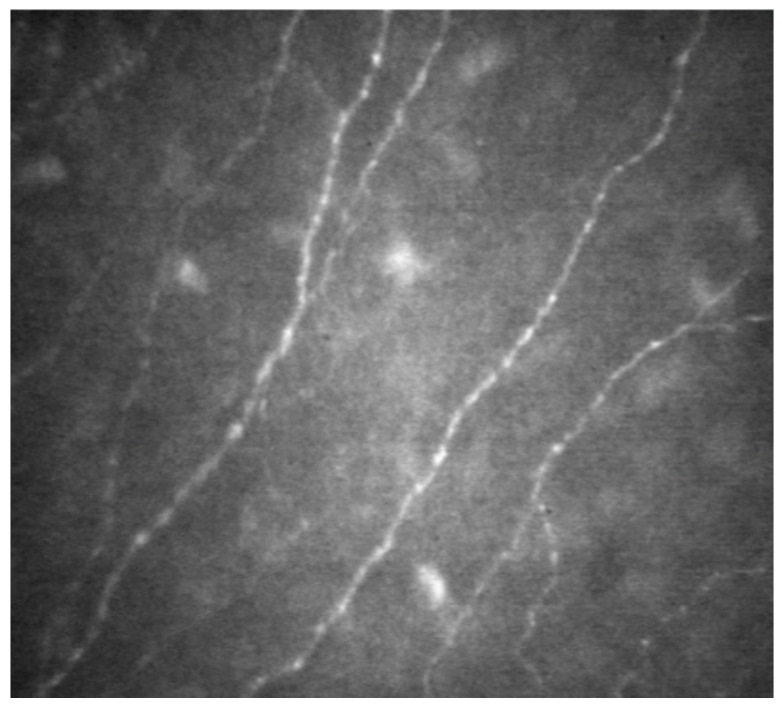
In vivo confocal microscopy (IVCM) visualization of subbasal nerve network. This image demonstrates the capability of IVCM in detailing the intricate network of nerves situated between the epithelial layer and the anterior stromal layer of the cornea, offering a clear view of their distribution and morphology.

## Data Availability

Not applicable.

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
