# Peer review of "Advances in Sjögren’s Syndrome Dry Eye Diagnostics: Biomarkers and Biomolecules beyond Clinical Symptoms"

_biomolecules, 2024, doi:10.3390/biom14010080_

Round 1

Reviewer 1 Report

Comments and Suggestions for Authors

Reviewer Comments (biomolecules-2800631-peer-review-v1)

The Review was overviewed the classification, pathophysiology, clinical features and presentation, ocular and systemic complications, clinical diagnosis, and management of Sjögren's syndrome dry eye (SSDE) to emphasized serum, saliva and tear samples. It was innovative in some ways, but there were some points to be need to clarifying I would like the authors to revise it.

1.      Figures (1 & 2) need to be interchanged among each other, similarly the references (e.g., 36 and 54 mismatch the year and incomplete) are not formatted according to the journal guidelines. The short form of abbreviations is not uniform throughout the manuscript. Revisions need to be made very carefully to this complete manuscript.

2.      The current review “Advances in Sjögren's Syndrome Dry Eye Diagnostics: Biomarkers and Biomolecules Beyond Clinical Symptoms” is very similar to his recently published review article from IJMS, (Wu KY, Kulbay M, Tanasescu C, Jiao B, Nguyen BH, Tran SD. An Overview of the Dry Eye Disease in Sjögren's Syndrome Using Our Current Molecular Understanding. Int J Mol Sci. 2023;24(2):1580. Published 2023 Jan 13. doi:10.3390/ijms24021580). So, I recommend to add some other new information, and their role between SSDE and NSSDE, like: SS key initiative factors and its chronic pathophysiology).

3.      SSDE subjects need to discuss relationships between 1. aquaporins, 2. cosmetic agents and 3. contact lens wearing in SS and DED. The authors pointed out that SS primarily affects women, so modern world needs to correlate with certain key errors.

4.      Somewhere spelling and words are incomplete (e.g. line-39, this paper; line-158, en-gendering; etc).

5.      Figure 4, anatomy of corneal layers images, is meaningless in this review. It is necessary for the authors to replace the advanced imaging tools (like, in vivo imaging etc.,,) with their output results from (3.4. Advanced Imaging Modalities).

6.      Section 4. A shortening or deletion is required for lines-292-305 of Novel Diagnostic Methods.

7.      Section 4.2.1. The lines 421 to 425 on salivary proteomics need to be removed or simply simplified.  

Author Response

Dear Reviewer,

Thank you for your constructive feedback on our manuscript. We appreciate the opportunity to enhance our work and address the concerns you have raised. Below, we have outlined how we have meticulously addressed each comment:

Figures and References Revision: We have interchanged Figures 1 and 2 as suggested. Additionally, we have meticulously reviewed and corrected the references, particularly 36 and 54, to align with the journal's formatting guidelines. The use of abbreviations throughout the manuscript has been standardized for consistency.

Concerning the second point you have raised (Distinguishing from Previous Work): We acknowledge your concern regarding the apparent similarity between our current review, "Advances in Sjögren's Syndrome Dry Eye Diagnostics: Biomarkers and Biomolecules Beyond Clinical Symptoms," and our previously published article in the International Journal of Molecular Sciences (IJMS). Upon reevaluation, we have further clarified in our manuscript how this review substantively diverges from our previous work.

While our previous article provided a comprehensive overview of Dry Eye Disease in Sjögren's Syndrome with a focus on molecular understanding, the current review emphasizes the cutting-edge diagnostic aspects, particularly the identification and application of biomarkers and biomolecules. The title of our current review, as you've pointedly observed, underscores this diagnostic slant, signifying a shift from a broad pathogenetic discussion to a targeted diagnostic approach.

In our revised manuscript, we now underscore the novel contributions of this review by elucidating the links between pathogenesis, pathophysiology, and the latest advancements in diagnostics, including a detailed examination of biomarkers and biomolecules. This revised narrative illuminates the unique and significant additions that this review makes to the literature on Sjögren's Syndrome Dry Eye (SSDE) diagnostics, which is distinct from our prior publication.

SSDE Subject Relationships: Thank you for your insightful comment regarding the exploration of relationships between aquaporins, cosmetic agents, and contact lens wearing within the context of Sjögren's Syndrome Dry Eye (SSDE) and Dry Eye Disease (DED) in our manuscript. Upon thorough examination of the literature, we have determined that, at present, there is a scarcity of high-evidence studies that explicitly correlate aquaporins, cosmetic agents, and contact lens usage directly with SSDE. While there is a body of research pertaining to these factors in the broader context of DED, these studies do not provide a robust or specific enough link to SSDE to warrant inclusion in our review according to our stringent analysis criteria. Our literature review was meticulous in searching for evidence that directly connects these factors with SSDE, but the available studies predominantly address DED in a general sense, without the specificity required for SSDE. Given the focus of our review on SSDE, we have chosen not to include these broader DED studies, as they do not meet the high-evidence threshold necessary to contribute meaningfully to our specific analysis. As such, we have refrained from discussing these relationships in the absence of substantial SSDE-specific data. We appreciate your understanding and hope that our rationale for this decision is clear.

Spelling and Grammar Corrections: We have thoroughly reviewed the manuscript and corrected all spelling and grammatical errors, including those specifically noted in lines 39 and 158.

Figure 4 Revision: We agree with your assessment of Figure 4. It has been replaced with images that reflect the results from advanced imaging modalities discussed in Section 3.4. This change provides more relevant and insightful visual information pertinent to our review.

Section 4 Revision: To enhance the clarity and conciseness of the manuscript, we have shortened lines 292-305 in Section 4, focusing specifically on the most novel and relevant diagnostic methods for SSDE.

Simplification of Section 4.2.1: The section on salivary proteomics (lines 421 to 425) has been simplified for clarity and brevity, ensuring that it contributes effectively to the overall narrative of the review.

We believe these revisions have substantially improved the manuscript and thank you for guiding these enhancements. We look forward to your further comments and suggestions.

Sincerely,

Reviewer 2 Report

Comments and Suggestions for Authors

I have read with great interest and attention this review about Sjögren's Syndrome Dry Eye. I found it to be a very interesting and very well prepared article where the authors collect and extensively discuss data previously described in the literature. Moreover, the figures are quite interesting.

I would like to congratulate the authors for their work.

Major changes:

A recent review on this syndrome was recently published in this editorial and I consider that it should be mentioned in this work. 1

-I would short the whole manuscript. It is too large and sometimes difficult to follow

Minor changes:

-when authors use an acronym for the first time (example SS, dry eye), they should use it in the rest of the manuscript

1- doi: 10.3390/life12111899. PMID: 36431034; PMCID: PMC9692499.

Author Response

Dear Reviewer,

We greatly appreciate your insightful comments and the time you have dedicated to reviewing our manuscript on Sjögren's Syndrome Dry Eye. Your positive feedback regarding the content and figures is highly encouraging. We are also grateful for your constructive suggestions, which we have thoroughly addressed as detailed below:

Major Changes:

  1. Inclusion of Recent Review: We acknowledge the importance of staying current with recent literature in the field. Following your suggestion, we have incorporated the mentioned review (DOI: 10.3390/life12111899) into our article (reference #45). This addition not only enriches our discussion but also ensures our review comprehensively covers the latest developments in the study of Sjögren's Syndrome Dry Eye. We have specifically referenced this work in sections where its relevance is most pronounced, providing a broader perspective on the topic.
  2. Manuscript Length and Clarity: In response to your concern about the manuscript's length and occasional complexity, we have undertaken a thorough revision to streamline the content. This involved shortening certain sections without compromising the comprehensive nature of the review, improving transitions between topics, and enhancing the overall clarity.

Minor Changes:

  1. Consistency in Acronym Usage: We appreciate your pointing out the inconsistent use of acronyms. We have revised the manuscript to ensure that each acronym (e.g., SS for Sjögren's Syndrome, DED for dry eye disease) is introduced clearly upon its first use and is then consistently utilized throughout the text. This change will aid in readability and avoid any potential confusion for the reader.

Thank you once again for your constructive critique and valuable guidance. We believe these modifications have significantly enhanced the quality and readability of our manuscript, aligning it more closely with the high standards of your journal.

Sincerely,

Round 2

Reviewer 1 Report

Comments and Suggestions for Authors

Overall, I accept this review as it stands. But it would be better if the authors displayed Section 3.4 (figure 4) to show SSDE and NSSDE in the discover overall representation previews of advance tools and diagnostic outcome images from IVCM, Meibography, and MRI/CT/US.

Reviewer 2 Report

Comments and Suggestions for Authors

Authors have addressed my comments. I know believe the article is more easy to read